Polyunsaturated fatty acids and their endocannabinoid-related metabolites activity at human TRPV1 and TRPA1 ion channels expressed in HEK-293 cells

Abate Atnaf atnaf.abate@students.mq.edu.au 1 2
Santiago Marina 1
Garcia-Bennett Alfonso 2 3
Connor Mark 1 2
1 Macquarie Medical School, Macquarie University , Sydney , NSW , Australia
2 Australian Research Council, Industrial Transformation Training Centre for Facilitated, Advancement of Australia’s Bioactives (FAAB) , Sydney , NSW , Australia
3 School of Natural Sciences, Macquarie University , Sydney , NSW , Australia
Gould Gwyn
Electronic publication date: 2025 Mar 24
Publication date: 2025
Volume: 13
Electronic Location ID: e19125
Received 2024 Oct 15; Accepted 2025 Feb 17
Copyright: ©2025 Abate et al.
Copyright year: 2025
Copyright holder: Abate et al.
License: This is an open access article distributed under the terms of the Creative Commons Attribution License, which permits unrestricted use, distribution, reproduction and adaptation in any medium and for any purpose provided that it is properly attributed. For attribution, the original author(s), title, publication source (PeerJ) and either DOI or URL of the article must be cited.
License URL: https://creativecommons.org/licenses/by/4.0/

Keywords: Cannabinoid, Fatty acid, Inflammation, Omega-3, TRP channel

Funding: Australian Research Council Industrial Transformation Training Centre for Facilitated Advancement of Australia’s Bioactives IC210100040 Office of the Chief Scientist and Engineer, Investment NSW Soho Floris International Switzerland SA, an industrial partner of the ARC, FAAB Soho Flordis International (SFI) (Australia) This work was supported by the Australian Research Council Industrial Transformation Training Centre for Facilitated Advancement of Australia’s Bioactives (Grant IC210100040) and the Research Attraction and Acceleration program funding from the office of the Chief Scientist and Engineer, Investment NSW. This study also received support from Soho Floris International Switzerland SA, an industrial partner of the ARC, FAAB, and by the Soho Flordis International (SFI) (Australia). There was no additional external funding received for this study. The funders had no role in study design, data collection and analysis, decision to publish, or preparation of the manuscript.

==============================
Background

Polyunsaturated fatty acids (PUFAs), particularly Omega-3 (ω-3) and Omega-6 (ω-6) PUFAs, may exert neuroprotective effects via the endocannabinoid system (ECS) and are promoted as brain health supplements. However, despite their potential role in endocannabinoid biosynthesis, the impact of PUFAs on ion channels such as TRPV1 and TRPA1, which are modulated by endocannabinoids, remains incompletely understood. Furthermore, the potential in vitro actions of ω-6 and ω-3 PUFA combined in the ratios available in supplements remains uncertain. Therefore, the objective of this study is to evaluate the functional activity of individual PUFAs, their combination in a specific ratio, and their endocannabinoid-related derivatives on TRPV1 and TRPA1 ion channels.

Methodology

We employed a fluorescent calcium-sensitive dye in HEK-293 Flp-In T-REx cells expressing human TRPV1, TRPA1, or an empty vector to measure changes in intracellular calcium concentration ([Ca]i).

Results

Capsaicin and PUFA derivatives such as docosahexaenoyl ethanolamide (DHEA), γ-linolenoyl ethanolamide (γ-LEA) and anandamide (AEA) stimulate TRPV1 activity directly, whereas eicosapentaenoic acid (EPA), docosahexaenoic acid (DHA), γ-linolenic acid (γ-LA), and their 9:3:1 ratio triggered TRPV1 response only after prior exposure to phorbol ester. Cinnamaldehyde and PUFA derivatives such as eicosapentaenoyl ethanolamide (EPEA), DHEA, γ-LEA, 2-arachidonoylglycerol (2-AG), 2-arachidonoylglycerol ether (2-AG ether) and AEA triggered TRPA1 response, with EPA, DHA, γ-LA, and the 9:3:1 ratio showing significant effects at higher concentrations.

Conclusions

PUFAs alone and their combined form in 9:3:1 ratio stimulate TRPA1 activity, whereas their metabolites trigger both TRPV1 and TRPA1 response. These findings suggest new avenues to explore for research into potential mechanisms underlying the neurological benefits of PUFAs and their metabolites.

INTRODUCTION

Polyunsaturated fatty acids (PUFAs) are crucial for maintaining brain function and have been used as supplements with claims of improving brain health (Bentsen, 2017; Bourre, 2009). In preclinical models, omega-3 (ω-3) and omega-6 (ω-6) PUFAs have demonstrated neuroprotective effects via the endocannabinoid system (ECS) (Dyall, 2017; Freitas et al., 2018).

The ECS is an essential part of the central nervous system (CNS) (Skaper & Di Marzo, 2012). Its core components include the lipid derivatives of PUFAs, endocannabinoids (eCB), enzymes regulating eCB synthesis and breakdown, and cannabinoid receptors CB1 and CB2 (DeMesa et al., 2021; Fezza et al., 2014; Lu & Mackie, 2016). Additionally, other G protein-coupled receptors like GPR55 (Lauckner et al., 2008; Yang, Zhou & Lehmann, 2016), GPR3, GPR119 (Davis, 2022) and GPR120 (Im, 2009) are potential member of the ECS. Peroxisome-proliferator activated receptors (PPARs) and transient receptor potential (TRP) ion channels, are also activated by various cannabinoid ligands, including eCBs (DeMesa et al., 2021; Lu & Mackie, 2016).

TRP channels are membrane proteins involved in sensing and responding to chemical and physical stimuli. They are integral to neural signalling processes related to various sensory perceptions including nociception (Julius, 2013; Muller, Morales & Reggio, 2019; Sawamura et al., 2017; Zhang et al., 2023). Specific channels within the TRP family, such as TRPV1, TRPV2, TRPV3, TRPV4, TRPA1, and TRPM8, have been identified as responsive to endogenous, phyto-, and synthetic cannabinoids (Muller, Morales & Reggio, 2019). It is also reported that these channels may contribute to eCB signalling, especially within the brain (Nilius & Owsianik, 2011). However, the potential modulation of these ion channels by ω-3 and ω-6 fatty acids derived eCBs has not been fully defined (Petermann et al., 2022).

Dietary PUFAs activate TRP channels, with eicosapentaenoic acid (EPA) and docosahexaenoic acid (DHA) shown to activate TRPV1(Ciardo & Ferrer-Montiel, 2017; Matta, Miyares & Ahern, 2007) and TRPA1 (Ciardo & Ferrer-Montiel, 2017; Motter & Ahern, 2012; Redmond et al., 2014) ion channels. These PUFAs also serve as major precursors for eCB biosynthesis (Komarnytsky et al., 2021), with resulting eCBs activating TRPV1 and TRPA1 ion channels. It has been reported that 2-arachidonoylglycerol (2-AG), anandamide (AEA) (Zygmunt et al., 2013; Zygmunt et al., 1999), and N-arachidonoyldopamine (NADA) (Huang et al., 2002; Raboune et al., 2014) activate TRPV1, while AEA activates TRPA1 (Redmond et al., 2014). However, there are also other ethanolamides or glycerol conjugates of PUFAs that may function as eCBs or ligands for related receptors; but their roles and targets in the brain remain unclear (Bosch-Bouju & Laye, 2016; Witkamp, 2016). The impact of the ω-6 fatty acid such as γ-linolenic acid (γ-LA) and its ethanolamine derivative on TRPV1 and TRPA1 ion channels has also not been examined.

Dietary intake affects brain PUFA levels (Dyall, 2017), and the ω-6 to ω-3 ratio in the current Western diet (approximately 20:1) is linked to various health conditions including autoimmune and inflammatory diseases (Patel et al., 2022; Simopoulos, 2002). One possible reason could be that ω-3 long-chain fatty acids such as EPA, DHA as well as some ω-6 derived fatty acids, e.g., γ-LA (particularly dihomo - γ-linolenic acid) are important for production of anti-inflammatory eicosanoids while ω-6 derived fatty acids, mainly arachidonic acid (AA), are crucial to produce pro-inflammatory eicosanoids. The presence of higher amounts of AA also interferes the synthesis of anti-inflammatory eicosanoids by competing at the active site of the enzyme cyclooxygenase (COX) (Calder, 2010). Therefore, to mitigate risks linked with excessive ω-6 PUFA consumption, maintaining a balanced ω-6 to ω-3 ratio between 1:1 and 5:1 has been suggested (Patel et al., 2022). Studies have also indicated that the ratio of ω-6 to ω-3 fatty acids in tissues is more important for health benefits than their absolute levels (Alshweki et al., 2015) and combining ω-6 and ω-3 PUFAs has shown positive health effects, especially in children with developmental coordination disorder (DCD) (Richardson & Montgomery, 2005). Moreover, it is noted that maintaining an equilibrium between ω-6 and ω-3 PUFAs in a healthy diet yields favourable effects on inflammation and other physiological mechanisms (Gomez-Candela, López & Viviana, 2011). Nevertheless, common agreement regarding the ideal ratio of dietary PUFAs remains elusive. Determining the optimum ratio of ω-3 PUFAs, EPA, and DHA for desired health benefits also remains uncertain (Djuricic & Calder, 2021; Gomez-Candela, López & Viviana, 2011; Mukhametove et al., 2022).

The objective of this study is to evaluate the individual PUFAs (EPA, DHA, and γ-LA), their combined form in a 9:3:1 ratio, and their endocannabinoid-related derivatives on the functional activity of human TRPV1 and TRPA1 ion channels, to develop a more complete picture of the mode of actions of these important dietary molecules.

METHODOLOGY

Portions of this text were previously published as part of a preprint (Abate et al., 2024).

Cell culture

Flp-In T-REx HEK-293 cells (Life Technologies, Mulgrave, Victoria, Australia), stably transfected with human TRPV1, TRPA1 cDNA (GenScript, Piscataway, NJ, USA) (Heblinski et al., 2020) or an empty pcDNA5/FR/TO vector (HEK 293-EV cells), were maintained in Dulbecco’s Modified Eagle’s Medium (DMEM) with 10% fetal bovine serum (FBS) (# 10099-141; Gibco), 100 U penicillin and 100 µg streptomycin ml−1 (1% P/S) (Gibco, # 15140-122, Life Technologies, USA), 80 µg ml−1 hygromycin B (cat # ant-hg-1; Invitrogen), and 15 µg ml−1 blasticidin (cat # ant-bl-1; Invitrogen, San Diego, CA, USA). The cells were incubated in a 5% CO2 humidified atmosphere at 37 °C. Upon reaching approximately 90% confluence in 75 cm2 flasks, they were trypsinized and transferred to poly-D-lysine coated 96-well plates (Corning, Castle Hill, NSW, Australia) in L-15 medium supplemented with 1% FBS, 100 U penicillin and 100 µg streptomycin ml−1 (1% P/S), and 15 mM glucose (80 µL volume per well). After an overnight incubation (maximum of 16 h) in humidified room air at 37 °C, TRPV1 and TRPA1 receptor expression was induced 4 h before experimentation by adding to each well 10 µL tetracycline solution to a final concentration of 1 µg ml−1.

Calcium assay

Intracellular calcium [Ca]i levels were assessed using the calcium 5 kit from Molecular Devices (# R8186, Sunnyvale, CA, USA) on a FlexStation 3 Microplate Reader (Molecular Devices, Sunnyvale, CA, USA). Calcium 5 dye was dissolved in Hank’s Balanced Salt Solution (HBSS) with the composition of (in mM): NaCl 145, CaCl2 1.26, MgCl2 0.493, HEPES 22, Na2HPO4 0.338, NaHCO3 4.17, KH2PO4 0.441, MgSO4 0.407 and glucose 1mg/ml (pH adjusted to 7.4, osmolarity = 315 ± 15 mOsmol) and used at 50% of the manufacturer’s suggested concentration. Probenecid (cat # 50027; Biotium) which helps to prevent expulsion of calcium indicator from the cells was added to a final concentration of 1.25 mM. 90 µL of the dye were loaded into each well of the plate and incubated for 1 h before reading in the FlexStation 3 at 37 °C. Fluorescence was recorded every 2 s (λ excitation = 485 nm, λ emission = 525 nm) for 5 min. After 1 min of baseline recording, 20 µL of the drug, dissolved in HBSS with 1% dimethyl sulfoxide (DMSO) (cat # D2650; Sigma-Aldrich, Melbourne, Australia) was added (final DMSO concentration was 0.1% in well). In order to demonstrate that HEK-293 EV cells responded with robust elevations of Ca in these experimental conditions, we used the protease activated receptor agonist PAR-1 to promote release of Ca from intracellular stores, via activation of endogenous receptors.

Drugs and reagents

All drugs were prepared in DMSO at a concentration of 30 mM and stored at −30 °C/−80°C. Freshly thawed aliquots were used in each experiment and diluted in HBSS containing 0.01% bovine serum albumin (BSA) (Sigma-Aldrich, # A7030). Due to limitations in the solubility of fatty acids and their derivatives, the highest concentration tested was 30 µM. EPA, DHA, γ-LA, and their endocannabinoid related derivatives and glycerol conjugates were procured from Cayman Chemical (Ann Arbor, MI, USA). Cinnamaldehyde (CA), a well characterized agonist of TRPA1 was obtained from Merck (Castle Hill, NSW, Australia). The 9:3:1 ratio were prepared by combining 9% of EPA, 3% of DHA and 1% of γ-LA dissolved in DMSO, so that, the highest concentration tested was 30:10:3 µM. In this study, we explored the effects of a 9:3:1 ratio of EPA, DHA, and γ-LA, which reflects the composition found in some commercially available omega-3 supplements. This ratio was chosen because an excess of EPA compared to γ-LA is prevalent in these supplements, designed to maximize the anti-inflammatory benefits associated with omega-3 fatty acids (Baker et al., 2016). All reagents for tissue culture were sourced from Merck or Life Technologies (Mulgrave, Victoria, Australia). Capsaicin, the canonical activator of TRPV1 (Caps) was obtained from Tocris Bioscience, Bristol (# 404-86-4) and the TRPV1 antagonist capsazepine was from Merck, USA (# 138977-28-3). PAR-1 agonist peptide (Thr-Phe-Leu-Leu-Arg-NH2, # 2660) was obtained from Auspep (Tullamarine, Victoria, Australia) while phorbol 12-myristate 13 acetate (PMA) was from Sigma-Aldrich, # P1585.

Data analysis

The response to agonists was expressed as a percentage change from the average baseline measurement taken for 60s before adding drug. Changes in fluorescence resulting from the addition of solvent were subtracted before normalization to baseline. Concentration–response curves (CRC), Emax and EC50 values were determined using a three-parameter logistic equation (GraphPad Prism, San Diego, CA, USA). Results are presented as the mean ± standard error of the mean (SEM) from at least six separate experiments conducted in duplicate, unless otherwise stated. Concentration–response curves of the positive controls for agonist activation, capsaicin and cinnamaldehyde, were obtained each day for comparative analysis. Potencies were expressed as pEC50, which is the negative log of EC50 in moles per litre (Molar). Where appropriate, unpaired Student’s t-test were used to compare the responses of individual compounds in different conditions, while a one-way ANOVA followed by Dunnett’s multiple comparisons test was used to assess potential differences in responses elicited by a range of compounds. P < 0.05 was considered statistically significant.

RESULTS

TRPV1 cellular responses to PUFAs and their bioactive metabolites

Application of docosahexaenoyl ethanolamide (DHEA), γ-LEA, AEA, 2- linoleoyl glycerol (2-LG), NADA, and capsaicin at 10 µM produced an elevation of [Ca]i in HEK-293 TRPV1 cells that was significantly greater than that produced in HEK-293 EV cells. However, EPA, DHA, γ-LA, their 9:3:1 ratio, as well as EPEA, 2-AG and 2-AG ether did not produce a notable change in [Ca]i in TRPV1 expressing cells, and there was no difference between the responses in HEK-293 TRPV1 cells and those seen in HEK-293 EV cells (Fig. 1; Table 1).

Figure 1 Representative traces of the response to PUFAs and their endocannabinoid related metabolites at 10 µM (A, B and C) in HEK-293 TRPV1 expressing cells.

Drugs were added for the duration of the bar. The representative traces for capsaicin are the same for each panel.

Table 1 Change of Ca5 fluorescence induced by PUFAs and their endocannabinoid related metabolites in HEK-293 EV and TRPV1 expressing cells.

Elevations of [Ca]i in response to PUFAs and their endocannabinoid-related molecules in HEK-293 EV cells, or cells expressing TRPV1. Drugs were tested at 10 µM or up to 30 µM, changes in [Ca]i are expressed as percentage of predrug baseline and were determined as outlined in the Methods. An unpaired t-test was conducted to compare their response in HEK-293 EV cell with that of HEK-293 TRPV1 expressing cells. An asterisk (*) indicates P < 0.05 compared to EV cells. The maximal percentage change in Ca5 fluorescence (Emax ± SEM) and potency (pEC50) for PUFAs and their derivatives in HEK-293 TRPV1 cells were reported either for the highest tested concentration (30 µM) or derived from concentration-response curves (DHEA, AEA and NADA). Capsaicin is included for comparison. All values represent the mean ± SEM of at least 6 determinations. “n.d” signifies “not determined”, while “-” denotes “not tested.”

DRUG	Mean ± SEM	E max ± SEM %	p EC 50 ± SEM	
	EV (10 µM)	TRPV1 (10 µM)	TRPV1 (30 µM)	TRPV1	
PUFAs and their metabolites	
EPA	14 ± 5	16 ± 5	34 ± 5	n.d	
DHA	11 ± 2	11 ± 2	29 ± 0.8	n.d	
γ - LA	9 ± 2	10 ± 1	40 ± 11	n.d	
9:3:1	9 ± 1	11 ± 2	38 ± 7	n.d	
EPEA	3 ± 1	15 ± 8	51 ± 22	n.d	
DHEA	10 ± 3	266 ± 51*	439 ± 81	5 ± 0.2	
γ-LEA	11 ± 2	277 ± 67*	354 ± 39	n.d	
AEA	10 ± 2	188 ± 36*	304 ± 55	5.1 ± 0.2	
2-AG	4 ± 2	30 ± 12	78 ± 26	n.d	
2-AG ether	8 ± 2	39 ± 24	63 ± 23	n.d	
2-LG	4 ± 1	41 ± 13*	75 ± 13	n.d	
NADA	6 ± 2	262 ± 40*	299 ± 45	5.5 ± 0.2	
Positive controls	
PAR-1 (100 µM)	463 ± 33	–	–	–	
Capsaicin	12 ± 2	350 ± 33*	340 ± 19	8.0 ± 0.1	

The effects of capsaicin and the active endocannabinoid-related metabolites of PUFAs in HEK-293 TRPV1 expressing cells were concentration-dependent. Capsaicin increased Ca5-dye fluorescence with a maximum response (Emax ± SEM) of 340 ± 19% above pre-drug and pEC50 ± SEM of 8.0 ± 0.1 (EC50 is expressed in molar units and derived from the data shown in Fig. 2). The PUFA-derived eCBs DHEA, γ-LEA, AEA, and NADA also triggered TRPV1 activation with the maximal responses that did not differ from those produced by the highest concentration of capsaicin (Table S1). However, despite the robust elevation of [Ca]i by these metabolites of PUFAs, we were unable to determine the potency (EC50 values) for these drugs because we could not construct complete CRCs for them, owing to the insolubility of these drugs at higher concentrations.

Figure 2 Concentration-response curves for endocannabinoid-related metabolites and capsaicin (A and B) in HEK-293 TRPV1 expressing cells.

The CRCs were fitted using a three-parameter logistic equation, each data point represents the mean ± SEM from at least six independent experiments conducted in duplicate.

EPA, DHA, γ-LA, and their combination in the ratio of 9:3:1 did not trigger TRPV1 responses (Fig. 3A). To test if they inhibited activation of TRPV1 ion channel, we applied a sub-maximally effective concentration of capsaicin (10 nM) after a 5 min exposure to EPA, DHA, γ-LA or their 9:3:1 ratio. Pre-treatment with EPA, DHA, γ-LA, or their 9:3:1 ratio (10 µM) did not affect the capsaicin response in HEK-293 TRPV1 expressing cells. Capsazepine (Bevan et al., 1992; Walpole et al., 1994) is an antagonist which was used for inhibition of capsaicin activation of the channel. Pre-incubation with capsazepine (10 µM) prevented capsaicin-induced fluorescence changes (P < 0.05) (Fig. 3B).

Figure 3 Concentration-response curves of PUFAs in HEK-293 TRPV1 expressing cells and effects of pre-incubation with PUFAs on responses to capsaicin in HEK-293 TRPV1 expressing cells.

(A) Concentration-response curves of PUFAs in HEK-293 TRPV1 expressing cells. The curves of DHA, γ-LA and in 9:3:1 ratio are overlapping; (B) effects of pre-incubation with PUFAs on responses to capsaicin in HEK-293 TRPV1 expressing cells. PUFAs or capsazepine (CAPZ) (10 µM each) were added to the cells for 5 min then capsaicin (10 nM) was added. Changes in [Ca]i are expressed as percentage of the pre-drug baseline. The black circles represent the response to first drug in individual experiments, while the pink circles correspond to the response to subsequent capsaicin addition. Two-way ANOVA with Dunnett’s multiple comparisons were used to analyze the response to capsaicin after pre-application of each PUFA and CAPZ, compared to capsaicin alone (HBSS served as a control in place of PUFAs and CAPZ). A significant difference (P < 0.0001) was observed only for capsaicin pre-treated with CAPZ, while no significant differences were detected for the other conditions.

TRPA1 cellular responses to PUFAs and their bioactive metabolites

Application of DHEA and γ-LEA at 10 µM as well as cinnamaldehyde at 300 µM produced an elevation [Ca]i in HEK-293 TRPA1 cells significantly greater than that produced in HEK-293 EV cells. However, at 10 µM, EPA, DHA, γ-LA, their 9:3:1 ratio, as well as EPEA, AEA, 2-AG and 2-AG ether did not produce a change in [Ca]i different to that seen in HEK-293 EV cells (Fig. 4; Table 2).

Figure 4 Representative traces of the response to PUFAs and their endocannabinoid related metabolites at 10 µM (A, B and C) in HEK-293 TRPA1 expressing cells.

Drugs were added for the duration of the bar. The representative traces for capsaicin are the same for each panel.

Table 2 Change of Ca5 fluorescence induced by PUFAs and their endocannabinoid related metabolites in HEK-293 EV and TRPA1 expressing cells.

Elevations of [Ca]i in response to PUFAs and their endocannabinoid-related molecules in HEK-293 EV cells, or cells expressing TRPA1. Drugs were tested at 10 µM or up to 30 µM, changes in [Ca]i are expressed a percentage of predrug baseline and were determined as outlined in the Methods. An unpaired t-test was conducted to compare their response in HEK-293 EV cell with that of HEK-293 TRPA1 expressing cells. The asterisk (*) indicates P < 0.05 compared to EV cells. The maximal percentage change in Ca5 fluorescence (Emax ± SEM) and potency (pEC50) for PUFAs and their derivatives in HEK-293 TRPA1 cells, was reported either for the highest tested concentration (30 µM) or derived from concentration-response curves. Cinnamaldehyde is included for comparison. All values represent the mean ± SEM of at least 6 determinations. “n.d” signifies “not determined”, while “-” denotes “not tested.

DRUG	Mean ± SEM	E max ± SEM %	p EC 50 ± SEM	
	EV (10 µM)	TRPA1 (10 µM)	TRPA1 (30 µM)	TRPA1	
PUFAs and their metabolites	
EPA	14 ± 5	58 ± 42	232 ± 67	n.d	
DHA	11 ± 2	129 ± 81	408 ± 54	n.d	
γ - LA	9 ± 2	112 ± 61	255 ± 73	n.d	
9:3:1	9 ± 1	85 ± 45	260 ± 61	n.d	
EPEA	3 ± 1	50 ± 40	274 ± 27	n.d	
DHEA	10 ± 3	107 ± 50*	335 ± 35	n.d	
γ-LEA	11 ± 2	87 ± 32*	311 ± 11	n.d	
AEA	10 ± 2	78 ± 40	206 ± 60	n.d	
2-AG	4 ± 2	6 ± 1	109 ± 68	n.d	
2-AG ether	8 ± 2	91 ± 40	223 ± 87	n.d	
2-LG	4 ± 1	6 ± 2	10 ± 4	n.d	
Positive controls	
PAR-1 (100 µM)	463 ± 33	–	–	–	
Cinnamaldehyde (300 µM)	9 ± 6	438 ± 29*	503 ± 23	4.4 ± 0.1	

Cinnamaldehyde, a commonly used activator of TRPA1, increased Ca5 fluorescence, with a maximum effect of 503 ± 23% above baseline with a pEC50 ± SEM of 4.4 ± 0.1. The highest concentration of EPEA, DHEA, γ-LEA, AEA, 2-AG, and 2-AG ether tested (30 µM), also activated TRPA1, while 2-LG had no effect (Table 2; Fig. 5). Among these metabolites, the response of EPEA, AEA, 2-AG, 2-AG ether, and 2-LG at 30 µM was significantly smaller than the response produced by the highest concentration of cinnamaldehyde (300 µM), while the maximal responses for DHEA and γ-LEA at the same concentration did not differ from that of cinnamaldehyde (Table S1). EPA, DHA, γ-LA, and their combination in a 9:3:1 ratio also activated TRPA1. At the highest concentration tested (30 µM), their maximum effects were 232 ± 67%, 408 ± 54%, 255 ± 73% and 260 ± 61%, respectively (Table 2; Fig. 6).

Figure 5 Concentration-response curves (CRC) for endocannabinoid-related metabolites and cinnamaldehyde (A and B) in HEK-293 TRPA1 expressing cells.

The CRCs were fitted using a three-parameter logistic equation, each data point represents the mean ± SEM from at least six independent experiments conducted in duplicate. CA represents cinnamaldehyde.

Figure 6 Concentration-response curves of PUFAs in HEK-293 TRPA1 expressing cells.

The CRCs were fitted using a three-parameter logistic equation from 6 separate experiments conducted in duplicate. CA represents Cinnamaldehyde.

PUFA-mediated TRPV1 activation following phorbol ester potentiation

Matta, Miyares & Ahern (2007) reported that some PUFAs, specifically EPA and DHA, activated rat TRPV1 ion channel only following incubation of cells with the phorbol ester protein kinase C activator, phorbol 12,13-dibutyrate (PDBu). Therefore, we assessed whether phorbol ester-dependent stimulation of human TRPV1 response by inactive PUFA could be observed. We initially determined the effect of PMA, another phorbol ester that is a protein kinase C activator (Jiang & Fleet, 2012) on HEK-293 TRPV1 expressing cells. Three concentrations of PMA were tested for 5 min followed by challenge with capsaicin at 10 nM (Fig. 7A). A total of 10 nM PMA had no effect by itself or on subsequent capsaicin responses, but 100 nM and 300 nM PMA increased [Ca]i by themselves and potentiated the subsequent response to capsaicin. However, to minimize potentially confounding effects, a concentration of 100 nM PMA was chosen. For these experiments, the response to PMA after 5 min was subtracted from the subsequent response to capsaicin or PUFA. PMA (100 nM) increased the effects of low, but not high, concentrations of capsaicin (Figs. 7B, 7C), presumably because of saturation of the TRPV1 channels at high agonist concentrations. The response of the tested PUFAs were also increased after exposure to PMA (P < 0.05 for each) (Fig. 7D). To rule out the possibility that the Ca5 dye was saturated at the highest concentration of capsaicin tested, we conducted experiments with ionomycin, a calcium ionophore (Kao, Li & Auston, 2010) (Fig. S2). Ionomycin (5 µM) produced a change in Ca5 fluorescence of 673 ± 25%, significantly greater than that produced by a maximal effective concentration of capsaicin (435 ± 15%).

Figure 7 PMA facilitates PUFAs activation of TRPV1.

(A) Response of HEK-293 TRPV1 expressing cells for PMA at various concentration. PMA were added to the cells at different concentrations for 5 min then capsaicin was added at 10 nM. One-way ANOVA with Dunnett’s multiple comparisons shows that 300 nM PMA potentiation significantly enhances the capsaicin response (P = 0.0098), while 100 nM and 10 nM PMA do not show a significant effect compared to capsaicin alone. (B) Traces of capsaicin and (C) Concentration response curves of capsaicin with and without PMA in HEK-293 TRPV1 expressing cells. Drugs were added for the duration of the bar; (D) The effects of PUFAs after potentiation of TRPV1 by PMA. The cells were pretreated with PMA (100 nM) for 5 min then each PUFA (10 µM) and capsaicin (10 nM) was added. Changes in [Ca]i are expressed as percentage of the pre-drug baseline, with the response to PMA alone at 5 min subtracted. The black circles represent the response of PUFAs without PMA pretreatment while the pink circles represent the response of PUFAs after PMA potentiation. Two-way ANOVA with Sidák’s multiple comparisons test revealed significant differences in the responses of EPA, DHA, and capsaicin before and after PMA potentiation, with p-values of 0.0396, 0.0059, and <0.0001, respectively.

DISCUSSION

This study demonstrates that derivatives of PUFAs, including DHEA, γ-LEA, and AEA, directly stimulate TRPV1, while EPEA, DHEA, γ-LEA, 2-AG, 2-AG ether, and AEA stimulate TRPA1. Moreover, EPA, DHA, γ-LA alone, and in a 9:3:1 ratio triggered TRPA1 activity directly, while stimulation of TRPV1 was only noted after PMA treatment of HEK-293 cells for EPA, DHA, γ-LA alone, and in a 9:3:1 ratio.

At 10 µM concentration, PUFAs and their endocannabinoid-related metabolites showed minimal effect on calcium levels in HEK-293 cells that did not express TRPV1 or TRPA1 (Fig. S1). Significant differences were observed when comparing their effect at the same concentration between EV cells and cells expressing TRPV1 (Table 1) or TRPA1 receptors (Table 2). At a concentration of 30 µM, most of the tested PUFAs and their metabolites elicited significant differences in responses between TRPV1 and TRPA1 expressing cells compared to EV cells at 10 µM (Tables 1; 2). These variations may be due to the expression of TRPV1 and TRPA1 or other cellular factors in these two cell types such as differences in signalling pathways, the presence of other receptors, or variations in the cell membrane properties that affect how the cells interact with or respond to the metabolites.

Previous studies have shown that TRPV1 responds to PUFA metabolites such as AEA (Zygmunt et al., 1999) and NADA (Huang et al., 2002) which aligns with the findings of our study. Our research has extended these findings by reporting that PUFA-derived eCBs such as DHEA and γ-LEA can trigger TRPV1 activation while EPEA showed minimal activity (Fig. 2; Table S2). Consistent with our findings, previous studies have shown that 2-AG is not very effective at activating these ion channels; but it is considered a physiologically relevant activator of TRPV1 channels through phospholipase C (PLC)-mediated mechanisms (Petrosino et al., 2016; Zygmunt et al., 2013; Zygmunt et al., 1999). Additionally, we showed that the related molecule 2-LG also minimally activate TRPV1 in these conditions.

Our findings indicate that the fatty acids EPA, DHA, γ-LA, and their combination in a 9:3:1 ratio did not trigger TRPV1 activation or inhibit its activity. As seen in earlier studies with rat TRPV1, phorbol esters enhance TRPV1 sensitivity to EPA and DHA through a PKC-dependent mechanism (Matta, Miyares & Ahern, 2007). In our experiments, this effect was confirmed at human TRPV1 as a significant response to EPA and DHA was only observed when a phorbol ester was present. Additionally, PMA amplified the TRPV1 response to capsaicin, γ-LA, and a 9:3:1 mixture of EPA, DHA, and γ-LA emphasizing the complex and context-dependent nature of TRPV1 activation. Matta, Miyares & Ahern (2007) also identified EPA as a competitive inhibitor of rat TRPV1; however, our experimental results do not support this finding. This difference could arise from variations in experimental conditions, such as our use of a calcium assay compared to their electrophysiology approach, differences in buffer composition, and order of application, all of which can influence channel activity and the efficacy of modulators. Specifically, the key difference between our study and Matta, Miyares & Ahern (2007) is the order of application. In our experiments, EPA was added before capsaicin, whereas in Matta, Miyares & Ahern (2007), capsaicin was applied before EPA. The order of application may influence the observed effects, as EPA might interact differently with the resting versus activated state of TRPV1. Pre-incubation with EPA, as in our study, may result in weaker inhibition if EPA binds less effectively to the resting conformation or is displaced by capsaicin upon its addition. Conversely, in Matta et al.’s study, EPA may have inhibited TRPV1 more effectively by interacting with the capsaicin-activated state. Further experiments testing different application orders are warranted to clarify this mechanism. Species-specific variations in TRPV1 structure and function may also contribute, as Matta, Miyares & Ahern (2007) used rat TRPV1, which may differ from human TRPV1 in terms of ligand sensitivity and binding affinity. Any of these factors may explain the absence of inhibition observed in our data.

Previous reports have shown that EPA, DHA, and AEA activate TRPA1 (Ciardo & Ferrer-Montiel, 2017; Motter & Ahern, 2012; Redmond et al., 2014) which aligns with our findings. We have extended this work to show that, PUFA-derived eCBs such as EPEA, DHEA, γ-LEA, and 2-AG ether can stimulate TRPA1 response, whereas 2-LG doesn’t activate it. TRPA1 activation plays a crucial role in various physiological processes, including protective mechanisms such as sensing pain and irritation, which help prevent further injury, and promoting immune cell recruitment as part of the inflammatory response. However, excessive or inappropriate activation of TRPA1, particularly in sensory neurons, can contribute to chronic pain and irritation (McNamara et al., 2007). The tissue-specific effects of TRPA1 activation remain an area of active investigation. TRPA1 receptors are expressed in the brain and other tissues (Nilius, Appendino & Owsianik, 2012), the physiological significance of PUFAs and their metabolites acting on this receptor are not yet fully understood, especially in the central nervous system. This uncertainty highlights the need for further research to determine the potential implications of TRPA1 activation by PUFAs in different tissues and contexts, particularly in the CNS, where its role remains poorly defined.

It has been reported that combining ω-6 and ω-3 PUFAs is important for achieving positive health outcomes compared to consuming these fatty acids individually (LaChance et al., 2016; Puri & Martins, 2014). Our results demonstrate that the combination of PUFAs activates the TRPA1 ion channel, suggesting a possible mechanism underlying some of these observed health benefits. However, since the physiological outcomes of TRPA1 activation may be complex, it cannot be generalized as it is inherently beneficial or detrimental.

We evaluated individual fatty acids such as EPA, DHA, and γ-LA and compared their effects to the combined PUFAs in the 9:3:1 ratio (30 µM with EPA at 20.77 µM, DHA at 6.92 µM, and γ-LA at 2.31 µM). The maximal activation by EPA and γ-LA (30 µM) was like that of the 9:3:1 combination, however, DHA at 30 µM appeared to be even more effective. Thus, the combination does not seem to be more effective in activating TRPA1 than the individual components. However, a combination of EPA, DHA, and γ-LA, with a higher proportion of DHA, may result in enhanced activation of TRPA1, which may have more profound effect in modulating neuroinflammatory processes. It is important to note that consuming a 9:3:1 ratio of ω-6 to ω-3 PUFAs does not necessarily result in the same ratio in tissues like the brain due to differences in absorption and metabolism. Future research should investigate how these fatty acids are distributed in specific tissues, including the CNS, and determine whether the combined effects of ω-6 and ω-3 PUFAs on TRPA1 are synergistic, additive, or simply reflect the sum of their individual actions. Our experiments did not explore and compare other possible combinations and ratios of PUFA which may form the focus of future studies.

CONCLUSION

The studied fatty acids stimulate the TRPA1 ion channel, while their metabolites trigger both TRPV1 and TRPA1 ion channels activity. Thus, local activation of these channels by PUFAs and their metabolites may influence neuronal function and provide positive effects through endocannabinoid-mediated mechanisms (Palazzo, Rossi & Maione, 2008). Our findings indicate that these dietary components could provide neuroprotective effects by modulating these ion channels. TRPV1 channel activation is beneficial for several neuronal functions, such as regulating synaptic plasticity, influencing cytoskeleton dynamics, and aiding in cell migration, neuronal survival, and the regeneration of damaged neurons. It also integrates various stimuli involved in neurogenesis and network integration (Marsch et al., 2007; Ramírez-Barrantes et al., 2016). TRPA1 activation is also associated with modulating inflammatory responses and neuropathic pain, thereby offering protection against neuronal damage and promoting overall brain health (Nassini et al., 2014). Generally, these results highlight the therapeutic potential of dietary PUFAs in influencing brain function through specific ion channel pathways, which could support neurological health and aid in preventing neurological diseases. However, while our study provides important insights into the potential effects of PUFAs on TRPV1 and TRPA1 ion channels, the mechanisms by which orally ingested PUFAs influence brain concentrations remain unclear. Therefore, future research should focus on elucidating these pathways to better understand how dietary intake of PUFAs through supplements might alter CNS levels of these critical compounds.

Supplemental Information

Supplemental Information 1 Comparison of the response of PUFA’s and their endocannabinoid related metabolites with the positive control of HEK-293 TRPV1 and TRPA1 expressing cells

* One-way ANOVA with Dunnett’s multiple comparisons was performed to compare the response of endocannabinoids and related metabolites of PUFAs at 30 µM with the highest concentration of capsaicin (10 µM) and cinnamaldehyde (300 µM) on HEK-293 TRPV1 and TRPA1 expressing cells, respectively. Unless specified otherwise, the sample size (N) is 6; a significance level of P < 0.05 indicates a statistically significant difference and those values which have significant difference are written in bold format.

Supplemental Information 2 PUFAs and their endocannabinoid metabolites included in this study, indicating previously studied metabolites and novel findings in HEK-293 TRPV1 and TRPA1 cells

Supplemental Information 3 Traces of PUFAs and their endocannabinoid related metabolites at 10 µM in HEK-293 empty vector cell

Supplemental Information 4 Traces of Ionomycin and Capsaicin in HEK-293 TRPV1 expressing cells

Supplemental Information 5 Raw Data

List of abbreviations

2-AG 2-arachidonoylglycerol

2-LG 2-linoleoyl glycerol

AA Arachidonic Acid

AEA Anandamide

CB1 Cannabinoid Receptor 1

CB2 Cannabinoid Receptor 2

DHA Docosahexaenoic acid

DHEA Docosahexaenoyl ethanolamide

DMEM Dulbecco’s Modified Eagle’s Medium

DMSO Dimethyl sulfoxide

eCB endocannabinoid

ECS Endocannabinoid system

EPA Eicosapentaenoic acid

EPEA Eicosapentaenoyl Ethanolamide

EV Empty Vector

FBS Fetal Bovine Serum

HBSS Hanks’ Balanced Salt Solution

HEK-293 Human Embryonic Kidney cell -293

NADA N-arachidonoyldopamine

P/S Penicillin-Streptomycin

PPARs Peroxisome-Proliferator Activated Receptors

PAR-1 Protease activated receptor 1

PUFA Polyunsaturated fatty acid

TRP Transient Receptor Potential

TRPA1 TRP ankyrin 1

TRPV1 TRP vanilloid 1

γ-LA γ-linolenic acid

γ-LEA γ-Linolenoyl ethanolamide

ω-3 Omega-3

ω-6 Omega-6

Additional Information and Declarations

Competing Interests

Author Contributions

Data Availability

The Soho Flordis International (SFI) (Australia) is a company that sells PUFAs for human consumption. Mark Connor is an Academic Editor for PeerJ. The authors declare there are no competing interests.

Atnaf Abate conceived and designed the experiments, performed the experiments, analyzed the data, prepared figures and/or tables, authored or reviewed drafts of the article, and approved the final draft.

Marina Santiago conceived and designed the experiments, authored or reviewed drafts of the article, and approved the final draft.

Alfonso Garcia-Bennett conceived and designed the experiments, authored or reviewed drafts of the article, and approved the final draft.

Mark Connor conceived and designed the experiments, analyzed the data, prepared figures and/or tables, authored or reviewed drafts of the article, and approved the final draft.

The following information was supplied regarding data availability:

Raw data can be found in the Supplemental Information.

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
