# Peer review of "Polyunsaturated fatty acids and their endocannabinoid-related metabolites activity at human TRPV1 and TRPA1 ion channels expressed in HEK-293 cells"

_PeerJ, doi:10.7717/peerj.19125_

## Round 0.1 · original submission · Major Revisions

While both reviewers recognise value in this study, both are clear that the presentation and organisation leave a lot to be desired. You therefore need to revise your paper in line with the specific suggestions and comments above, and make sure than in the revised version you produce a clear narrative flow to the work so that readers can understand what has been done, what is being shown and why you have done it. The idea of a summary cartoon-style figure should be adopted.

Note that this should not be regarded as a quick fix issue. The manuscript needs a complete overhaul and should be carefully edited for narrative flow, logic and clarity.

Reviewer 1 ·

Basic reporting

In this manuscript, Abate et al. explored the effects of polyunsaturated fatty acids (PUFAs), including Omega-3 and Omega-6 fatty acids, and their endocannabinoid-related metabolites on TRPV1 and TRPA1 ion channels, recognized for their neuroprotective roles. Using HEK-293 cells expressing TRPV1 and TRPA1, the authors measured calcium influx to analyze channel activation (calcium assay). Findings indicated that some PUFA derivatives, like DHEA and AEA, directly activated TRPV1, while other PUFAs required pre-exposure to phorbol ester to achieve channel stimulation. Additionally, metabolites such as EPEA and 2-AG activated TRPA1 channels. In general, these results may highlight the therapeutic potential of PUFAs dietary in affecting brain function through specific ion channel pathways, which could aid in preventing neurological diseases. Moreover, as the authors suggested in the paper, further research should be conducted to understand the consumption of the 9:3:1 ratio of Omega-6 to Omega-3 PUFAs and how they are distributed in the nervous system; it is also important to consider where these PUFAs and their metabolites are most likely to exert their effects in the body.
Comments to be considered:
1. Divide the results into subsections with clear, descriptive titles summarizing the main findings.
2. Line 192 (“Its potency (pEC50±SEM) was 8.0 ± 0.1”) should include concentration units. Also, clarify if “8.0” represents EC50 derived from Figure 1D and E, as this is currently unclear.
3. Provide a table listing the PUFAs and endocannabinoid-related metabolites used in the study for easier reference.
4. Replace “GLA” with “ɣ-LA” throughout the manuscript (lines 158,294), including in Figures 2, 4, and 5.
5. In Figure 2A, clarify if DHA and GLA data are combined with other lines and update the legend accordingly. For Figure 2B, match color descriptions in the legend with the actual colors used in the figure (black and pink instead of blue and red), indicate the statistical significance (for each group, and clarify for which group the p<0.0001 is assigned), and specify the statistical test applied.
6. Define “positive control” consistently across the manuscript. Lines 189-190 – “The effects of the positive control and endocannabinoid-related metabolites of PUFAís in HEK190 293 TRPV1 expressing cells were concentration-dependent,” where it is shown that the positive control shows concentration dependency unless the positive control is capsaicin and cinnamaldehyde then you should clarify and change the definition of the Capsazepine in line 207 as a “positive control.”
7. Figure 5B is not mentioned in the manuscript, specifying concentrations used for capsaicin (e.g., low concentrations, etc.) to clarify interpretation.
8. Figures 5A and D: Present statistical significance in Figures 5A and 5D, mentioning the test used. Moreover, in lines 241-242, “The response of the tested PUFAs were also increased after exposure to PMA (P<0.05 for each) (Figure 5D), meaning that the increase after the exposure is significant in the same degree for all the compounds compared to the drugs alone? Please explain and provide the exact statistical significance for each group.
9. Include a reference to Supplementary Figure 1 in the text, in the discussion regarding the empty vector in lines 255-256. Clarify the purpose of the PAR-1 evaluation in the empty vector, as its relevance to the main findings is currently unclear.
10. In lines 258-260, cite the relevant figures or tables showing the response to increased PUFA concentration (Tables 1 and 2? Because Table 1 presents the mean and Table 2 presents the Emax). Consider adding a table comparing concentration effects for easier visualization.
11. Lines 203-210: you showed that “EPA and DHA did not affect the capsaicin response in HEK-293 TRPV1 expressing cells,” although Matta et al. 2007 indicated that “Among n-3 PUFAs, EPA and LNA are the most effective antagonists of capsaicin-evoked currents, whereas DHA does not possess antagonistic properties. Low micromolar concentrations of LNA and EPA inhibited TRPV1 responses in HEK 293 cells (at room temperature and 37°C) and in sensory neurons. Furthermore, in animals, EPA markedly reduced capsaicin-evoked pain-related behaviour, whereas DHA was considerably less effective. “ so, EPA still has inhibition probabilities as a competitive inhibitor.

Experimental design

NA

Validity of the findings

NA

Additional comments

NA

Reviewer 2 ·

Basic reporting

Major concerns are that it took a series of readings to grasp what the data were telling me because the organization of the results section is confusing and lacks structure. There are so many complex results, but the authors provide a few paragraphs of general reporting and references to Tables and Figures that are difficult to follow. For example, Table 1 has a clearer explanation in the legend though it is confusing to have both TRPV1 and TRPA1 in the same table and then have all the examples of traces divided up in Figures and skip back and forth to try to understand what was being referred to. I had to read and reread the narrative about Table 2 to understand that it was (if I understand correctly) the same type of data using the same cell types and ligands, but at 30uM dose? If that is correct, it needs to be more clearly described, if it is incorrect, it also needs to be more clearly described. Regardless of what Table 2 is representing, it would be more informative to have the two doses using one cell type on a single Table.
The clarity and readability of Figures 1-5 is poor, especially 1, 3, and 5. It may be an issue of transforming to PDF, but the font is quite small and the lines and color blur making some of the traces very difficult to discern. There is also the issue of the use of the same colors for different compounds. For example, AEA is the same color as EPA, EPEA, and NADA all in the same figure, just different panels. It is extremely confusing. In addition, AEA is represented twice in the same figure (understandable), but it is 2 different colors?!
The organization of the results also lends to the confusion where it is written more as a rambling narrative instead of distinct units of information. Again, if the authors would present the data from TRPV1 and then the data for TRPA1 separately and then use headings and sub-headings, the readers would be able to follow the details more accurately.

The lack of organization and focus extends to the discussion, which would benefit greatly from being presented in sections. The lead sentence, “The current study demonstrates that some derivatives of PUFAs directly stimulate the TRPV1 and TRPA1.” is an example of the lack of clarity and focus and is statement that could be the lead sentence in the introduction as the fact that “some derivatives of PUFAs…” have been known to activate these channels for decades. The authors then switch between these types of sentences and the use of strings of acronyms for the individual ligands that can create confusion in those not accustomed to these terms.
Some of the paragraphs in the discussion are more informative than others but read as if they were pasted in from another narrative and make it difficult to follow. As an example, the paragraph that starts on line 266 starts with the sentence, “Transient receptor potential vanilloid subtype 1 (TRPV1), which is widely recognized as the capsaicin receptor (Caterina et al. 1997), has previously been reported to be responsive to metabolites of PUFAs such as AEA (Zygmunt et al. 1999) and NADA (Huang et al. 2002) corroborating our findings.” Why do the authors define TRPV1 again in the discussion and restate this information from the introduction after we have been reading TRPV1 for 16 pages? While clearly not a major issue (redefining an acronym), it speaks to the disjointed nature of the narrative and the likelihood that this paragraph was lifted from some other narrative and pasted there without thought for how it helped the reader understand this complex data set.
Ideally, there would be the addition of a summary cartoon type figure with TRPV1 ligands that were novel on one side and TRPA1 on the other to help the reader have a better grasp of the overall findings.

Minor
PMA is missing from the drug list in the Methods.

Experimental design

Acceptable though clarifications of the results will help to understand the overall design.

Validity of the findings

The manuscript, “Polyunsaturated fatty acids and their endocannabinoid-related metabolites activity at human TRPV1 and TRPA1 ion channels expressed in HEK-293 cells.” by Abate and colleagues provides important information about key TRP channel members and how their responses are regulated by endogenous lipids and those used as dietary supplements, such as PUFAs. It is important to add the information of the field that PUFAs on their own do not activate TPV1 channels in any appreciable way but that they do activate TRPA1.

---

## Round 0.2 · accepted · Accept

Thanks for the clear and professional way in which you have responded to the suggestions of the referee and I. I am delighted to recommend acceptance now.

Reviewer 2 ·

Basic reporting

The authors have revised the report in a way that makes the concepts, hypotheses, methods, and results more clearly defined. These revisions highlight the strength of the study and its importance to the field.

Experimental design

Clearly defined, understandable, and validated.

Validity of the findings

Data are very clear and convincing.

Additional comments

Authors addressed all previous concerns and the manuscript is a solid example of well-executed studies and analyses.